# Friends or Foes—Microbial Interactions in Nature

**DOI:** 10.3390/biology10060496

**Published:** 2021-06-02

**Authors:** Nancy Weiland-Bräuer

**Affiliations:** Institute of General Microbiology, Kiel University, 24118 Kiel, Germany; nweiland@ifam.uni-kiel.de; Tel.: +49-431-880-1649

**Keywords:** microorganisms, interaction, symbiosis, metaorganism, metagenomics, biofilms, quorum sensing, quorum quenching

## Abstract

**Simple Summary:**

Microorganisms like bacteria, archaea, fungi, microalgae, and viruses mostly form complex interactive networks within the ecosystem rather than existing as single planktonic cells. Interactions among microorganisms occur between the same species, with different species, or even among entirely different genera, families, or even domains. These interactions occur after environmental sensing, followed by converting those signals to molecular and genetic information, including many mechanisms and classes of molecules. Comprehensive studies on microbial interactions disclose key strategies of microbes to colonize and establish in a variety of different environments. Knowledge of the mechanisms involved in the microbial interactions is essential to understand the ecological impact of microbes and the development of dysbioses. It might be the key to exploit strategies and specific agents against different facing challenges, such as chronic and infectious diseases, hunger crisis, pollution, and sustainability.

**Abstract:**

Microorganisms are present in nearly every niche on Earth and mainly do not exist solely but form communities of single or mixed species. Within such microbial populations and between the microbes and a eukaryotic host, various microbial interactions take place in an ever-changing environment. Those microbial interactions are crucial for a successful establishment and maintenance of a microbial population. The basic unit of interaction is the gene expression of each organism in this community in response to biotic or abiotic stimuli. Differential gene expression is responsible for producing exchangeable molecules involved in the interactions, ultimately leading to community behavior. Cooperative and competitive interactions within bacterial communities and between the associated bacteria and the host are the focus of this review, emphasizing microbial cell–cell communication (quorum sensing). Further, metagenomics is discussed as a helpful tool to analyze the complex genomic information of microbial communities and the functional role of different microbes within a community and to identify novel biomolecules for biotechnological applications.

## 1. Introduction—Microbial Interactions at a Glance

Microorganisms, or short microbes, are speciesism of microscopic scale, including the highly diverse group of unicellular organisms belonging to the three domains of life, comprising bacteria, archaea, protozoa, microalgae, fungi, and viruses. Microorganisms are present in nearly every niche on Earth and their global distribution is striking, ranging from the human gut to deep subsurface in terrestrial and marine environments and the upper atmosphere [1]. Prokaryotes (bacteria, archaea) and viruses form the majority of microorganisms and, consequently, represent the review’s focus. Exemplarily, bacteria reach abundances of 1 × 10^8^ cells/g and viruses even of 5 × 10^9^ particles/g in dry soil; in oceans, bacteria achieve densities of 5 × 10^5^ cells/mL and viral particles 1 × 10^11^ viruses/mL [2]. Microbes mainly do not exist solely but form communities of single or mixed species [3]. Within such microbial populations, and between the microbes and a eukaryotic host or the environment, a huge variety of microbial interactions occur, ranging from bacteria–bacteria, bacteria–fungus, bacteria–virus, to bacteria–host (plant, animal) interactions [4].

Those microbial interactions are crucial for successfully establishing and maintaining a microbial population in various environments and on various hosts [5]. The many years of coevolution of the different species led to interdependent adaptation and specialization and resulted in various symbiotic relationships facilitating commensal, mutualistic, and parasitic interactions [5] (as illustrated in Table 1). Mutualism describes a win-win situation for both partners, such as for *Rhizobium* spp., which colonizes the plant’s roots to fix nitrogen in exchange for nutrients [6]. Moreover, human gut bacteria synthesize the essential vitamin K in the lower gastrointestinal tract to exchange carbon sources [7]. Relationships in which only one partner benefits, but the other is not affected, is defined as commensalism, in contrast to parasitism, in which the benefiting partner harms the other partner. The evidence is increasing that commensal bacteria, which reside in the human gut and airways, profoundly affect the regulation of immunophysiological functions, including metabolism, ontogeny, and pathogen defense [8,9]. Contrarily, parasitic bacteria, better known as pathogens, harm their host in various ways, such as invading tissues, producing toxins, or causing direct damage to host cells. Pathogens causing diseases are well studied and broadly recognized by the public, such as *Bacillus anthracis*, the cause of anthrax [10]; *Borrelia* spp., the cause of Lyme disease [11]; *Campylobacter jejuni*, a cause of gastroenteritis [12], and *Haemophilus influenza*, an agent of bacterial meningitis and respiratory tract infections [13]. The list of parasitic interactions, particularly focusing on human diseases, could be expanded indefinitely; however, mechanistic understanding of commensal and mutualistic interactions, especially between prokaryotes, lags [14]. Microbial interactions occur through the transfer of molecular and genetic information, including the exchange of secondary metabolites, signaling molecules, cellular transduction signals, siderophores, or genetic elements [5]. The basic unit of the interaction is the gene expression of each organism in response to biotic or abiotic stimuli, responsible for producing exchangeable molecules involved in these interactions [15].

In the following, examples of microorganism–microorganism and microorganism–host interactions are presented in more detail to demonstrate the variety and diversity of microbial interactions in different habitats. 

## 2. Microbial Community Interactions

Microbes respond to their chemical environment and interact with other microbes in their vicinity [22]. The nature and significance of interactions depend on the abundance and types of microorganisms present, which possess different sensory systems [23]. Cell–cell interactions can cause cooperative effects, where one or more individuals benefit, or competition between microbes occurs with an adverse effect on one or more partners. Microbes are not limited to a single type of interaction, and their response is transient and influenced by the chemical and/or physical environment, resulting in a highly complex microbial community [5].

### 2.1. Fungal Interactions

Fungi inhabit a broad range of environmental niches and account for at least 25% of the global biomass. In their natural environment, fungi interact with other microorganisms, such as other fungi and bacteria (as illustrated in Figure 1). Both intra- and interspecific fungal interactions are mediated upon contact and/or signaling molecules leading to, e.g., mating, alterations in growth and development, and pathogenicity [24]. For instance, *Burkholderia* acts as an endosymbiotic partner of *Rhizopus microspores*, causing rice seedling blight disease. The bacterium produces the rice-killing toxin rhizoxin and enables the fungus to produce infecting spores [25]. Additionally, fungi must compete with other organisms and among each other for resources, such as nutrients and space. Competition occurs, for instance, by secretion of secondary metabolites or by direct interaction, such as overgrowth and mycoparasitism [26]. Mycoparasitism, where one fungus attacks and invades another, significantly contributes to the suppression of pathogens. Mycoparasite *Trichoderma* is therefore applied in the biological control of fungal plant diseases [27]. Moreover, it was shown that fungal–bacterial interactions enable the production of specific fungal secondary metabolites [28]. It was demonstrated that exclusively close physical interaction between *Aspergillus nidulans* and *Streptomyces rapamycinicus* activates the production of specific aromatic polyketides [29]. The actinomycete thus triggers alterations in histone acetylation to affect fungal gene regulation [30]. In general, Actinomycetes are producers of many natural products with a wide range of bioactivities [31]. A study on soil-dwelling *Streptomyces coelicolor* interacting with other Actinobacteria showed that most of the compounds produced in each interaction were unique for the respective partnership. Many novel bioactive molecules and an extended family of acyl-desferrioxamine siderophores were identified. In total, over 200 differentially synthesized compounds were identified, including prodiginines possessing immunosuppressive and anticancer activities, actinorhodin antibiotics, and siderophores coelichelin and acyl-desferrioxamines [32]. Thus, Actinobacteria interspecies interactions seem to be very specific and complex, and harbor enormous potential to identify novel biotechnologically and medically relevant compounds [31]. 

Within the last decades, it became clear that fungal interactions are crucial to natural and anthropogenic ecosystems, including human health. On the one hand, fungal interactions represent a great potential to be utilized in sustainable agriculture. It is frequently suggested that arbuscular mycorrhizal may improve phosphor nutrition, enhance nitrogen uptake, or improve disease resistance in their host plants. Other microbes, e.g., nitrogen-fixing bacteria or phosphor-solubilizing bacteria, synergistically interact with those fungi and benefit plant development and growth [33]. The mycorrhizal symbiosis becomes important in sustainable agricultural systems where nutrient inputs are low and play an essential role in nutrient mobilization from crop residues [34].

On the other hand, recognizing fungal interactions with harmful properties, for instance, in human health, could lead to improved therapeutics [35].

### 2.2. Archaeal Interactions

Although archaea may be the most ancient organisms on Earth [37], only recently researchers became aware of the multiple ways in which archaea may interact with each other and with organisms of other kingdoms (as illustrated in Figure 2). In particular, methanogens take part in essential steps of the global methane cycle, which are partially conducted in a symbiotic interaction with different partners, such as herbivorous animals or sulfate-reducing bacteria [38]. No archaeal pathogen was identified until now, though some archaeal commensals may be indirectly involved in bacterial infections [39]. Mutualistic symbioses with archaeal partners were described, some of them with high relevance to global environmental cycles and others of unknown ecological significance related to highly specific mechanisms (reviewed in [38]). However, the identification and study of archaeal interactions are challenging because interactions between predominantly uncultivable or at least complicated cultivable organisms are problematic to detect [40]. Nevertheless, the archaeal symbiosis between the host *Ignicoccus hospitalis* and *Nanoarchaeum equitans* is well described even at the structural level [41]. *Ignicoccus* (Crenarchaeota, Desulfurococcales) is an anaerobic, hyperthermophilic obligate chemolithoautotrophic hydrogen oxidizing archaeon. The symbiont *N. equitans* directly attaches to the specialized outer membrane of *Ignicoccus* and obligatorily depends on the *Ignicoccus* host because the highly reduced genome lacks genes for essential biosynthetic pathways, such as lipid, amino acid, and nucleotide biosynthesis [42]. Consequently, biological macromolecules must be provided by *Ignicoccus* [43] (as illustrated in Figure 2). Another study observed stable archaeal aggregates formed by *Pyrococcus furiosus* and *Methanopyrus kandleri*, while hydrogen produced by *Pyrococcus* is utilized by *Methanopyrus* [44]. Such interspecies hydrogen transfer is also prominent for syntrophic archaea–bacteria consortia. For example, the consortium of “*Methanobacillus omelianskii*” comprises a methanogenic archaeon and a Gram-negative bacterium, which in syntrophy convert ethanol to acetate and methane [45,46]. A multitude of such syntrophic associations was described for hydrogenotrophic methanogens, for example, with the fermentative *Acetobacterium* and *Syntrophobacter* [47,48], *Desulfovibrio* under low sulfate concentrations [49], and *Thermoanaerobacter*, *Desulfotomaculum*, and *Pelotomaculum* under thermophilic conditions [50,51,52]. In addition, an essential process of methane oxidation in anoxic sediments is conducted by consortia of Euryarchaeota (ANME, anaerobic methanotroph) and sulfate-reducing bacteria (SRB), like *Desulfovibrio* and *Desulfococcus*. The partners often form small aggregates up to voluminous mats [53] (as illustrated in Figure 2). Recently, cell–cell interaction between the giant filamentous *Thaumarchaeote candidatus, Giganthauma karukerense,* and a sulfur-oxidizing Gamma-Proteobacterium was described. Here, the bacteria build a monolayer, which covers the surface of the large *Thaumarchaeote* filament and most likely reduces the sulfide concentration around the host cell [54].

Furthermore, methanogenic archaea are essential in the degradation of organic substrates under anaerobic conditions to methane and carbon dioxide within the guts of animals as anaerobic niches of nutrient decomposition [55] (as illustrated in Figure 2). Remarkably, a single methanogenic representative, *Methanobrevibacter smithii,* is the predominant archaeon in human gut microflora [56]. A syntrophic interaction between *Methanobrevibacter* and *Bacteroides thetaiotaomicron*, as studied in gnotobiotic mice, may affect the energy balance of the host. *Methanobrevibacter* utilizes the *Bacteroides* fermentation product formate. This syntrophy further determines the expression of *Bacteroides* enzymes [57]. Symbioses between archaea and eukaryotes, however, are not restricted to the gut anaerobic food chain. Symbioses between methanogens and protists are also well known (as illustrated in Figure 2). Here, the archaea are directly attached to the hydrogenosomes of anaerobic protozoa and ciliates [58]. Archaea are also ubiquitous in marine sponges, sometimes even dominant [59], though their ecological role is almost unknown, but were recently correlated to sponge nitrogen metabolism [60,61,62] (as illustrated in Figure 2). Interactions between archaea and other organisms are definitely as specific and widespread as bacterial interactions, but so far, the underlying mechanisms are still poorly understood. The findings mentioned above give just an impression on the abundance and diversity of archaeal interactions. Future studies with archaeal model organisms might lead to insights comparable with that of those achieved with bacterial models like *E. coli* and *Pseudomonas aeruginosa* [38].

### 2.3. Virus–Bacteria Interactions

Viral interactions are ecologically fundamental since viruses are responsible for many diseases in various eu- and prokaryotic hosts [63]. The present review focuses on the viruses infecting bacteria and the modulation of bacterial communities through those (bacterio) phages. Phages are the most abundant microorganisms in the biosphere, with an estimated 4.8 × 10^31^ particles [64]. Phages are present in all areas, coexisting parallel to their bacterial hosts in the majority of ecosystems [65]. Phages are well known as drivers of microbial diversity, vectors of horizontal gene transfer, sources of diagnostic and genetic tools, and therapeutics [65]. Bacteriophages are natural killers of bacteria and can be classified according to their genome, morphology, biological cycle, or provenance [66]. A prominent distinguishing feature of phages is their biological cycle [67]. There are two main types, the lytic and lysogenic cycle (as illustrated in Figure 3). During the lytic cycle, phages attach to the host cell’s surface to inject their nucleic acids into the cell. Subsequently, the host cell DNA is degraded, and the host metabolism is directed to initiate phage biosynthesis. The phage nucleic acids replicate inside the bacterium. Accordingly, entire viral phage particles are assembled, which are released from the infected cell through lysis of the bacterial cell. In contrast, the lysogenic cycle is based on the integration of the genetic material of the phage into the genome of the host cell to produce a prophage (or temperate phage). When the bacterium reproduces, the prophage is also copied and passed to each daughter cell. The daughter cells can continue to replicate with the prophage, or the prophage can exit the bacterial chromosome to initiate the lytic cycle [68]. The initiation of phage infection is triggered by the specific recognition between the phage-binding protein located at the tip of the tail or the capsid envelope and a receptor located on the host cell’s surface [69]. Cell surface receptors recognized by the phage may include protein receptors (OmpA and OmpC), lipopolysaccharide (LPS) receptors, receptors located in capsular polysaccharides (Vi-antigen), and pili and flagella [70]. 

Interactions with phages can cause benefit or harm to individual cells or entire communities because interactions range from mutualistic to commensal up to parasitic, including the transmission of novel bacterial phenotypes, modulation of bacterial gene expression and evolution, and killing of bacteria [71]. Some temperate phages can modulate bacterial physiology, such as *E. coli* phage Mu, which integrates randomly within the bacterial genome, mutagenizing an infected population and eliminating those cells with insertions in essential genes [72]. Lytic phages modulate microbial communities by simply lysing infected cells. There is increasing evidence from aquatic habitats that phages massively affect bacterial diversity, bacterial virulence, bacterial evolvability, and even shape the stability of ecosystems [73]. Phages also alter the structure and function of microbial communities via horizontal gene transfer mediated by generalized transduction and transformation [74]. Prophages enhance bacterial defenses against invading phages by blocking almost every step of viral infection and replication, including adsorption, DNA injection, transcription, or even induction of altruistic suicide of the hosts. Moreover, phages modulate eukaryotic physiology [71]. Phage-conditioned changes in microbial community compositions were associated with various diseases, including Crohn’s disease and ulcerative colitis (expansion in phage richness) and type I diabetes (diminished phage richness and diversity) [75]. Bacteriophage research nowadays increasingly focuses on the potential of phages to treat bacterial infections and contaminations [76,77]. The worldwide increase of pathogenic bacteria resistant to antibiotics requires alternative strategies to combat this threat [78]. Phage therapy has a tradition dating back almost a century, but further development slowed down in Western countries when antibiotics were discovered [77]. The therapeutic use of bacteriophages is one promising strategy in the medical field, food industry, and agri- and aquaculture. Many attributes of phages suggest a positive outcome in therapy, both by preventing contaminations and treating ongoing infections, particularly biofilm infections [79]. As demonstrated, phage–bacteria interactions are manifold, but their study, particularly for interactions with archaea and fungi, lags. Similarly, less knowledge is available about phage–phage interactions, which can affect the course of bacteria–phage interactions, with implications for the microbial community and associated multicellular organisms. Novel technologies, such as sequencing viromes from nature, imaging of viruses, and genetic engineering for a molecular understanding of the underlying interactions, direct future research and ultimately enable gaining insights into those fundamental relationships [65].

### 2.4. Bacteria–Bacteria Interactions

Interactions between bacteria, either between cells of the same species or between different bacterial species, are manifold and ubiquitous in nature [23] (as illustrated in Figure 4). Bacterial interactions got into focus by case studies on communication and cooperative behavior among myxobacteria, Quorum Sensing (QS), and biofilm formation of *Pseudomonas* [80]. Recently, cooperative behaviors among bacteria were increasingly considered in evolutionary biology [4]. Extracellular signal molecules produced by bacterial cells can be detected by other cells and regulate the expression of genes. In some bacteria, siderophores are synthesized for microbial community interactions [23,81]. In *Pseudomonas* spp., pyoverdine siderophores are essential for infection and biofilm formation, and may potentially help to regulate bacterial growth [82]. In the marine environment, exogenous siderophores act as signaling compounds that influence the growth of marine bacteria under iron-limited conditions. Many marine bacteria were reported to produce iron-regulated outer membrane proteins exclusively in the presence of exogenous siderophores produced by other species, such as *N*, *N*’-bis (2,3-dihydroxy benzoyl)-*O*-serylserine from *Vibrio* sp. [83]. Various cooperative and competitive bacterial interactions rely on synthesizing and detecting small chemical signaling molecules in a communication process called QS (QS). Several social behaviors of bacteria are triggered or affected by QS, such as virulence, pathogenicity, and biofilm formation [84]. Although the molecular structures of the signaling molecules, the organization of the sensing machinery, and the functional consequences of the signaling process show significant diversity among different bacteria, the biological similarity of these processes is undeniable [85]. In the following, biofilms are examined in more detail as the most common form of syntrophic microbial consortium in nature. Moreover, QS as a fundamental and universal communication strategy of bacteria within the domain of bacteria and among different domains is considered and QS-regulated behaviors are highlighted. Finally, interference with bacterial cell–cell communication (Quorum Quenching, QQ) is discussed as a natural mechanism for recycling own QS signals and in the context of a competitive relationship. Further, QQ is considered a natural occurring strategy to prevent and inhibit colonization of (antibiotic-resistant) bacteria, particularly those embedded in a biofilm, with promising potential for future applications.

#### 2.4.1. Biofilms

Biofilms are one of the most widely distributed and successful modes of microbial life [86]. A biofilm is defined as an aggregate of microorganisms in which cells are embedded in a self-produced matrix of extracellular polymeric substances (EPS) [87]. The interaction of cells initiates biofilm formation with a surface or with each other. It is supposed that the planktonic bacteria adhere to the surface initially through reversible adhesion via van der Waals forces. Attached cells proliferate and produce an extracellular matrix to form microcolonies, in which communication among the cells through biochemical signals and a genetic exchange is facilitated. The matrix contains exopolysaccharides, extracellular DNA, RNA, and proteins. Cells further proliferate, and spatial structuring occurs in all dimensions, resulting in a mature three-dimensional biofilm. Over time, microcolonies undergo cell death and lysis along with active dispersal of motile bacteria [88] (as illustrated in Figure 5). A biofilm is assumed to maintain an equilibrium through growth and dispersal [89]. Biofilms are complex systems typically comprising many species of high cell densities, ranging from 10^8^ to 10^11^ cells/g wet weight [90]. Biofilms drive biogeochemical cycling processes of most elements in water, soil, sediment, and subsurface environments. All higher organisms are colonized by biofilms, which can be correlated with persistent infections in plants and animals, including humans [1]. Furthermore, biofilms cause contamination of medical devices and implants, biofouling, contamination of process water or even drinking water, and corrosion [91]. In contrast, biofilms are used in biotechnological applications, including filtration of drinking water, degradation of waste (water), and biocatalysis of biotechnological processes, such as producing bulk and fine chemicals as well as biofuels [92]. 

The biofilm lifestyle is distinct from that of free-living bacterial cells. Biofilm communities have emergent properties, like physical, metabolic, and social interactions, including enhanced gene exchange and increased tolerance to antimicrobials [93]. Tolerance in biofilms can result from both the biofilm matrix acting as diffusion barrier and inactivation zone of antimicrobials and slowed growth of biofilm cells, even leading to the dormancy of cells. Further, a high proportion of stationary cells (persisters) with changed metabolic performance were detected in biofilms [93]. Resistance of cells in the biofilm to antimicrobials can also occur by the uptake of resistance genes through horizontal gene transfer, since genetic competence and accumulation of mobile genetic elements is increased in biofilms [94]. The organization in biofilms allows and promotes interactions for a myriad of organisms due to the created proximity. Proximity enables the exchange of metabolites, signaling molecules, and genetic material between organisms [86] (as illustrated in Figure 4). Furthermore, heterogeneity, such as cells with different metabolic capacities or physiological gradients, provides opportunities for cooperation [95]. The heterogeneous physiological activity in biofilms produces vertical gradients of electron acceptors and donors, pH value, and redox conditions [96]. One of the most important external triggers of the establishment of gradients is the availability of electron acceptors such as oxygen, resulting in aerobic microcolonies in the upper layer of the biofilm and the formation of anaerobic zones in deep layers [97]. Heterogeneity in biofilms also enables spatial organization of mixed species such as in microbial mats. Here, phototrophic microorganisms, e.g., algae, cyanobacteria, and anoxygenic phototrophic bacteria generate and release organic substrates as exudates, which are used from neighboring heterotrophic species in close proximity, thus enhancing their metabolic activity [98] (as illustrated in Figure 4). Metabolic interactions between different species in biofilms can also be observed in the process of nitrification, in which ammonia-oxidizing bacteria convert ammonium into nitrite, which is subsequently oxidized by nitrite-oxidizing bacteria [18] (as illustrated in Figure 4). Examples of cometabolism or metabolic sharing lead to more efficient resource partition between community members, further supporting the concept of coevolution of biofilm members [86]. Cooperation does not necessarily occur in all biofilms, and it was even suggested that most species–species interactions in biofilms are negative, since cells are competing with each other [23]. Competition mechanisms in biofilms include antibiotics, bacteriocins, extracellular membrane vesicles, and type VI secretion systems. This ultimately causes inhibition of initial adhesion of the biofilm, surface colonization (e.g., swimming and swarming of *P. aeruginosa* cells on the surface, thereby preventing the adhesion of competing *Agrobacterium tumefaciens* cells [99]), or the production of biosurfactants with antimicrobial properties [100] (as illustrated in Figure 4). Furthermore, invaders can inhibit the maturation of a biofilm and promote its dispersal through downregulation of adhesin synthesis, inhibition of cell–cell communication, or degradation of matrix polysaccharides, nucleic acids, and proteins [101].

In conclusion, numerous biofilm studies identified fundamental principles that underlie many of the key properties and phenotypes of biofilms, e.g., cell–cell interactions, spatial structuring, and heterogeneity. Although studies of microbial consortia in natural settings were revolutionized by metagenomics, most insights were gained with less complex biofilm communities in the laboratory, often neglecting spatial and temporal scales of microbial interactions in the assemblages [3]. Understanding how to disrupt or promote the function of biofilm communities, which are recognized as the primary form of bacterial life in nature, is a priority for modern microbiology [95]. Consequently, extensive knowledge gain on QS, which plays a crucial role in biofilm formation for various bacterial species, is essential and would have immense implications for an improved understanding of microbial ecology and the treatment of microbial infections. 

#### 2.4.2. Bacterial Communication—Quorum Sensing

Many bacteria use a cell–cell communication system called Quorum Sensing to coordinate population density-dependent behaviors [80]. QS is based on the synthesis and perception of low molecular weight molecules, so-called autoinducers (AI), which either diffuse over the cytoplasmic membrane or are actively transported and detected explicitly by a specific receptor (as illustrated in Figure 6). When the AI binds its corresponding receptor, the subsequent signal transduction is activating the transcription of target genes, often including those encoding the respective AI synthase (autoregulation) [102]. When the population density increases, the concentration of the signaling molecule is passing a threshold (“quorum”), thus causing more autoinducers to be synthesized through the induction of AI synthase. This forms a positive feedback loop, and the receptor becomes fully activated. Activation of the receptor changes the regulation of target genes, leading to synchronized transcription in the population [103]. Thus, cell density-dependent behaviors are coordinated (for review, see [104]), e.g., colonization, virulence, pathogenicity, and biofilm formation as mentioned above [105,106]. 

QS systems were found in both Gram-negative and Gram-positive bacteria [107]. Gram-negative bacteria communicate via acylated homoserine lactones (AHLs) (as illustrated in Figure 6A) [104]. The first description of QS was on the bioluminescent marine bacterium *Vibrio fischeri*. At high cell densities, in symbiotic association with the Hawaiian bobtail squid *Euprymna scolopes*, *V. fischeri* activates bioluminescence through QS and supports the squid in masking its shadow during predator avoidance. Two components, LuxI and LuxR, impact the expression of target genes, e.g., the *lux* operon (*luxICDABE*) responsible for bioluminescence in *V. fischeri* [108]. Similar AHL QS systems were since shown to be widely distributed in Gram-negative bacteria controlling diverse behaviors, such as the production of secreted toxins and virulence factors, biofilm formation, and conjugation. Many bacteria harbor more than one signal-receptor combination [109]. For example, *P. aeruginosa* has two complete LuxRI-type homologs, LasRI and RhlRI, which operate in a hierarchy [110]; and *B. thailandensis* has even three LuxRI homologs [111]. It was proposed that energy-costly resourcing of different AHL communication systems might provide specific benefits in different environments [112]. Gram-positive bacteria communicate using modified oligopeptides and two-component regulatory systems (as illustrated in Figure 6B). Briefly, the signaling molecules are either unmodified or posttranslationally modified small peptides secreted via ABC exporter proteins. Phosphorylation of the receptor kinase due to peptide binding activates the regulatory protein, which acts as a QS target gene transcription factor [113]. Virulence factor production in *Bacillus cereus* and *Staphylococcus aureus*, the competence of *B. subtilis*, and the biofilm formation of Streptococcus pneumonia are only a few examples of QS-dependent gene regulation in Gram-positive bacteria [114,115]. QS allows bacteria not only to communicate within their own, but also between different bacterial species. Therefore, autoinducer-2 (AI-2) is synthesized and recognized by many different bacterial species. Thus AI-2 appears to be an almost universal signal (as illustrated in Figure 6C) [116]. The AI-2 system was first described in *V.**harveyi* [117]. The AI-2 synthase, called LuxS, produces the AI-2 precursor, 4,5-dihydroxy-2,3-pentadione (DPD) [117]. DPD can spontaneously cyclize to generate some isoforms, collectively referred to as AI-2 [118,119,120]. Different isoforms bind different signal receptors; for instance, the S-form binds to the signal receptor LuxP in *V. harveyi*, whereas the R-form binds to the LsrB receptor protein in *Salmonella enterica serovar Typhimurium* or *E. coli* [121]. For instance, it was reported that AI-2 is involved in the regulation of bacteriocin production and biofilm formation in *S. mutans*, biofilm formation of *S. anginosus* and *Listeria monocytogenes*, virulence regulation of *S. pneumonia* and *S. pyogenes*, and toxin production in *Clostridium* (reviewed in [122]). In general, the bacterial response to certain autoinducers is manifold and adaptable. Vastly different bacterial genera can detect the same compound as in the case of AI-2. Slightly modified molecules of the same chemical class even activate different responses among different species of the same genus [123]. Correspondingly, a number of chemically different QS molecules act jointly in a particular organism [124].

Recent studies further focused on the role of QS in cooperative and competitive microbial interactions, thus concentrating on QS as a social behavior [80]. Many QS-regulated products are secreted or excreted products, such as secreted proteases, and can thus be used by any community member, although its synthesis implies a metabolic cost for only one individual cell [125]. QS-dependent cooperation was for instance demonstrated for regulated production of elastase in *P. aeruginosa*, a protease required for growth when populations are cultivated on casein as the sole source of carbon and energy [126,127]. In addition, bacterial swarming is a social trait due to the joint production of secreted surfactants in several bacterial species, including *P. aeruginosa* and *B. subtilis* [128,129,130]. In contrast, various bacterial species use QS to control the production of secreted or cell-targeted toxins, for example, bacteriocins in *Streptococcus* species [131,132] and type VI secretion effectors in *B. thailandensis* [133]. In soil communities, *P. fluorescens* and *P. aureofaciens* use QS-regulated phenazines to fight the fungus *Gaeumannomyces graminis* and colonize the plant [134]. AHL-dependent competition was likewise observed for *P. aeruginosa* and *S. aureus* mixed communities, which commonly coculture in chronic wound infections. *P. aeruginosa* usually surpasses or decreases the *S. aureus* population by QS-regulated synthesis of compounds, which block *S. aureus* oxidative respiration, such as 4-hydroxy-2-heptylquinoline *N*-oxide or pyocyanin. Subsequently, *P. aeruginosa* induces *S. aureus* cell lysis by the QS-regulated protease LasA [135]. 

Moreover, recent evidence shows that QS is not restricted to the domain of bacteria, but also shows that QS is not restricted to bacteria and allows communication between bacteria and their hosts. In the meantime, scientists comprehended that these bacterial signals modulate mammalian cell signal transduction [136], and that host hormones can crosstalk with QS signals to modulate bacterial gene expression [137]. These observations are not surprising since prokaryotes and eukaryotes coexisted for millions of years, and the development of eukaryotes depended on bacterial communities [138]. The research field of “interkingdom signaling” is still in its infancy, but the increasing number of publications in this area demonstrates that microbial–host communication is in the spotlight. Prominent examples of microbial–host communication are presented in the following, which disclose their importance for bacteria–host interactions. The first example of QS between bacteria and plants was found in the relationship between the marine bacterium *V. anguillarum* and the green seaweed *Enteromorpha* [139]. Biofilm-forming *Vibrio* cells release AHLs and attract zoospores, the motile reproductive stage of the seaweed, which subsequently settle to establish and develop in a certain habitat. In the meantime, several QS-dependent seaweed-microbe interactions are known (for review, see [140]). The first demonstration of a specific response of a plant to bacterial AHLs was shown for the legumes *Phaseolus vulgaris* [141] and *Medicago truncatula* [142]. Here, AHLs from both symbiotic (*Sinorhizobium meliloti*) and pathogenic (*P. aeruginosa*) bacteria caused significant changes in the plants’ gene expression. In addition, Gao et al. have shown that *M. truncatula* responds to bacterial communication by producing its small molecule AHL-mimics [143]. The most studied plant–bacteria interaction in the marine environment is the red alga *Delisea pulchra*, which secretes brominated furanones to protect from fouling microorganisms. The algae release those brominated furanones, which inhibit multiple AHL-dependent processes, including swarming motility in *Serratia liquefaciens* and bioluminescence in *Vibrio* spp. as well as AI-2-based QS in *Vibrio* spp. [144,145]. Due to their QS inhibitory effects, halogenated furanones showed practical potential for treating disease in shrimp aquaculture and reduced the virulence of *P. aeruginosa* in mouse models [146]. Moreover, one of the best-studied interkingdom signaling mechanisms with plant hosts is the relationship between *Rhizobium* spp. and their symbiotic legume host. In this symbiosis, complex exchange of signals between bacteria and the plant leads to the successful formation of root nodules, in which bacteria reside and fix atmospheric nitrogen [147]. Plants evolved multiple mechanisms to interpret bacterial QS signals and initiate attraction/defense responses in a tissue-specific manner, which are even signal-specific [142,148]. Fungi also communicate by small signaling molecules and even talk to bacteria in their vicinity [149]. However, fungi were not shown to produce bacterial autoinducer analogs [150]. The most prominent example is the yeast *Candida albicans*, whose QS molecule farnesol acts in a cell density-dependent manner and causes a morphological switch between yeast and hyphae [151]. Since the discovery of farnesol, QS was described in several other fungal species and was shown to be involved in regulating growth, stress resistance, morphogenesis, and biofilm formation [149,152]. So far, identified fungal QS molecules include peptides, e.g., of *Cryptococcus neoformans* [153], oxylipins in *A. nidulans* [154], and alcohols and their derivatives such as tyrosol in *C. albicans* [155]. Furthermore, there is increasing experimental evidence that bacteria can recognize mammalian hormones. Here, research mostly focuses on pathogens, which turn on their production of virulence factors to respond to mammalian hormones. The AI-3/epinephrine/norepinephrine signaling system is a prime example. The enteric pathogen *E. coli* senses AI-3 produced by the microbial gastrointestinal (GI) community to activate virulence genes resulting in colon lesions. Eukaryotic hormones epinephrine and norepinephrine present in the GI tract activate the expression of the virulence genes in enterohemorrhagic *E. coli* (EHEC) [137]. Thus, EHEC captures the eukaryotic hormones, subsequently promoting colonization of the human colon mucosa that causes colon lesions. Such an adrenergic regulation of virulence seems not to be restricted to EHEC. In silico analyses observed this form of interaction in other bacterial species such as *Salmonella spec.*, *Shigella flexneri*, *Francisella tularensis*, *H. influenzae*, *Erwinia carotovoa*, *Pasteurella multocida*, *Ralstonia eutropha*, *Chromobacterium violaceum,* and *V. parahaemolyticus* [156,157]. In addition, opioids such as endorphin and dynorphin are known as novel hormones hijacked by pathogenic bacteria like *P. aeruginosa*. The bacteria recognize those opioids to enhance their virulence by increasing the production of their QS systems leading to persistent *P. aeruginosa* colonization in the lungs of cystic fibrosis patients [158]. Besides the mentioned involvement of AIs in bacterial pathogenesis, likewise the effects of AIs on eukaryotic cells should not be dismissed. There is increasing evidence that AIs, i.e., 3-oxo-C12 homoserine lactone (HSL), are able to modulate signal transduction and immune responses of the eukaryotic host [159]. In addition, high concentrations of 3-oxo-C12-HSL induced apoptosis due to calcium mobilization from the endoplasmic reticulum [160]. In contrast to the detrimental effects caused by the QS signals of *P. aeruginosa*, a study by Fujiya et al. suggests a cooperative bacteria–host relationship mediated by bacterial QS [161]. Gram-positive *B. subtilis* synthesize a pentapeptide (competence and sporulation factor, CSF) to regulate expression of competence and sporulation. Nevertheless, CSF also activates two crucial kinase-dependent survival pathways in intestinal epithelial cells by preventing cell injury and loss of barrier function [161].

The numerous examples of QS involvement within the bacterial domain and among domains leading to cooperative and competitive interactions point to the importance of this fundamental communication system. Since many bacteria use QS to control the expression of virulence factors, regulate pathogenicity and biofilm formation, the interference with this cell–cell communication mechanism further constitutes a novel and promising strategy to control bacterial infectious diseases [162,163,164,165].

#### 2.4.3. Interference with Bacterial Communication–Quorum Quenching

The term “Quorum Quenching” (QQ) describes all processes that interfere with bacterial cell–cell communication [166]. In a polymicrobial community, some bacteria are communicating with neighboring cells by QS, while others are interrupting the communication due to QQ mechanisms; thus, long time thought of as primarily operating as a defense mechanism against competitors [84,90,167]. The different QQ mechanisms operate by blocking different steps involved in QS, comprising (i) blocking signal generation and accumulation, (ii) preventing signal reception, and (iii) inhibiting autoinduction and activation [109] (as illustrated in Figure 6D). (i) Firstly, inhibition of signal molecule biosynthesis can be achieved by inhibiting involved enzymes as the acyl chain (acyl-acyl carrier protein) (ACP) and S-adenosylmethionine synthase, or interfering with the synthases themselves as LuxI homologs and LuxS [165]. (ii) Secondly, several small molecules that mimic or deactivate the complex interactions between the signaling molecule and their protein receptors were identified [168]. A distinction is made between an agonist, implying a function like the native AHL based on the mimic structure and an antagonist blocking the receptor binding site prevents binding of the signaling molecule [169]. For instance, an AHL agonist for *P. aeruginosa* was identified, which shows no obvious structural connection to the AHL (3-oxo-C12-HSL) but was predicted to bind in the same protein pocket of the receptor protein as the AHL [170]. In addition, a series of naturally occurring bromo-furanones exhibit potent antagonistic QS inhibition and appear to function by disturbing the dimerization of the receptor protein and not by competitive binding at the ligand site [171]. Further, diketopiperazines (DKP) are cyclic dipeptides, which share structural similarities to signaling peptides in mammalian tissues. They are produced by various bacteria such as *P. aeruginosa*, *Proteus mirabilis*, *Citrobacter freundii,* and *Enterobacter agglomerans* [172], and yeast, fungi, and lichens [173]. DKP act as AHL antagonists in LuxR-based QS and as agonists in others [172]. Also, cross-inhibition by autoinducing peptides (AIPs) of Gram-positive bacteria in *S. aureus* represents an example of QQ mechanism by inhibitors because each of the four AIPs present in *S. aureus* specifically inhibits QS in competitive *S. aureus* groups [174]. Each AIP specifically activates its cognate receptor but inhibits activation of all others by competitive binding to the noncognate receptors. Thus, each AIP inhibits activation of the virulence cascade in the other three groups of *S. aureus*. Coinfection with two different *S. aureus* groups results in intraspecies competition; the *S. aureus* group that first establishes its QS cascade suppresses the other group [175]. Based on the importance of S-ribosyl-homocysteine (SRH) in synthesizing the precursor DPD for the generation of universal signaling molecule AI-2, several research groups found substrate analogs of SRH potential inhibitors that target AI-2 synthesis. SRH analogs, S-anhydroribosyl-L-homocysteine and S-homoribosyl-L-cysteine, exhibited inhibitory activities against LuxS [176]. Several SRH analogs were further reported as potential LuxS inhibitors. Kinetic studies indicate that these compounds act as reversible, competitive inhibitors against LuxS [177]. As SRH is the most important intermediate for the synthesis of DPD, 5-methylthioadenosine nucleosidase (MTAN) is also important as an enzyme during the synthesis process. MTAN is encoded by the *pfs* gene in bacteria and catalyzes the hydrolytic deadenylation of its substrates to form adenine and S-ribosylhomocysteine. According to the mechanism of the reaction catalyzed by MTAN, several transition state analogs, e.g., But-DADMe-ImmA, were designed and synthesized, inhibiting AI-2 synthesis [178]. Moreover, several agonist ligands were reported for *V. harveyi* receptor protein LuxP, most of which are DPD or AI-2 (S-THMF-borate) analogs competing for binding to LuxP with natural AI-2 [179,180]. Ren et al. found that the natural furanone compound (5Z)-4-bromo-5-(bromomethylene)-3-butyl-2(5H)-furanone could inhibit the AI-2-mediated QS in *V. harveyi* and *E. coli* [181]. A screening of many samples from plants, ursolic acid, and 7-hydroxy indole was found as inhibitors for enterohemorrhagic *E. coli* biofilms by blocking the AI-2 pathway [182,183]. Previous research showed that certain food components inhibit AI-2 signaling using reporter strain *V. harveyi* BB170 [184]. AI-2 QS inhibitors in poultry meat wash samples were characterized by identifying several quenching fatty acids. Linoleic acid, oleic acid, palmitic acid, and stearic acid expressed AI-2 inhibition ranging from approximately 25–99% [185]. (iii) Thirdly, modification or degradation of the QS signaling molecules prevent them from accumulating. QS signal degradation can be mediated by chemical, metabolic, and enzymatic mechanisms [186]. The chemical degradation was reported primarily at alkaline pH, leading to the opening of the lactone ring of AHLs [187]. However, at acidic pH, the ring recyclizes, and the activity is restored. A few organisms such as *Variovorax paradoxus* and *P. aeruginosa* can metabolize AHLs as the sole carbon source, thus suppressing competitive QS bacteria parallel to energy generation [188,189]. Enzymatic degradation of QS signaling molecules was observed in a wide range of prokaryotes and eukaryotes. AHL-lactonases hydrolyze the ester bond of the homoserine lactone ring of AHL molecules [190]. The first reported AHL-lactonase encoded by the *aiiA* gene was characterized from *Bacillus isolate* 240B1 [191]. Homologs were identified in a range of bacteria, including Gram-positive and Gram-negative species. AHL-lactonases can be grouped into two clusters based on their sequence homologies. The first one is the AiiA cluster with representatives from *Bacillus* [192]. The second one is the AttM cluster with Gram-negative members, e.g., *A. tumefaciens* and *Klebsiella pneumoniae* [193,194]. AHL-lactonases are by far the most specific AHL-degrading enzymes among known QQ enzymes. They hydrolyze both short- and long-chain AHLs but show no residue activity to other small molecules [195]. Paraoxonases (PON) were identified in mammals, other vertebrates, and invertebrates [196,197] and are also capable of hydrolyzing the homoserine lactone ring of AHLs [173]. PON enzymes seem to be most active with long-chain AHL molecules, often used by eukaryotic pathogens, e.g., *P. aeruginosa* [198]. PONs are well known for their broad-spectrum enzyme activities, unlike lactonases [199]. AHL-acylases inactivate AHL signals by cleaving the amide bond of AHL, thus producing the corresponding fatty acids and homoserine lactone [188]. These enzymes are widely conserved in several bacteria, including *Variovorax*, *Ralstonia,* and *P. aeruginosa* [200,201]. There are notable differences in the substrate specificities among AHL-acylases, which are demonstrated in the effectiveness of degrading long-chain AHLs [202]. AHL-oxidoreductases do not degrade AHLs but modify the 3-oxo group of the molecule to generate corresponding 3-hydroxy derivates [203]. Depending on the specificity of the AHL receptor, the modification may or may not affect the signaling activity of the respective AHL [204]. Quenching of Gram-positive signaling is, for instance, enabled by NADPH oxidases located in the membranes of phagocytes, which are responsible for the generation of bactericidal reactive oxygen species during host defense and essential for the innate immune system [205]. The enzyme inactivates autoinducing peptides (AIP) through its enzymatic products [206]. For instance, the inactivation of the AIP signal of *S. aureus* is caused by oxidation of the C-terminal methionine sulfanyl group of the signal to the corresponding sulfoxide form resulting in the loss of AIP activity [207]. In contrast to various AHL-quenching mechanisms and compounds, only very few AI-2 interfering mechanisms, in particular AI-2 QQ enzymes, were reported so far. *E. coli* AI-2 kinase LsrK was used in vitro to phosphorylate AI-2, resulting in reduced QS response when added ex vivo to *E. coli* populations as well as *Salmonella typhimurium* and *V. harveyi* cultures [208]. Highly effective inhibition of AI-2 regulated biofilm formation of *Klebsiella* spp. was demonstrated by the first metagenome-derived AI-2 Quenching enzyme. Here, AI-2 signals were most likely modified by the identified oxidoreductase QQ-2 [209].

QQ is considered as a natural mechanism evolved either by organisms regulating behaviors via QS for the recycling or clearing of their QS signals or by QQ organisms in the context of a competitive relationship with QS signal-producing organisms [166]. QQ was found to be related to the fine-tuning of QS functions, e.g., clearing of the QS-signal regulated the transfer of the Ti-plasmid in *A. tumefaciens*, which is crucial for plant infection with crown gall disease [210]. Recycling of QS signals mainly occurs in microorganisms that produce QS molecules [166]. For example, in *P. aeruginosa,* the amidase HacB and PvdQ, and QuiP contribute to AHL recycling by converting AHLs into fatty acids and homoserine lactone (HSL) further assimilated by the bacterium [211]. Likewise, LsrF and LsrG are involved in the degradation of AI-2, thereby terminating the induction of the *lsr* operon and closing the AI-2 signaling cycle in *E. coli* [212]. In microorganisms that do not produce QS but are sensitive to the toxicity of QS signals, QQ enzymes play a significant role in detoxification [213]. In Gram-positive *Bacillus* strains, which communicate via peptides and not AHLs, the AiiA lactonase was identified. *Bacillus* spp. protect themselves from AHLs by degrading those signaling molecules, which express a bactericidal activity against several Gram-positives [166,214]. Moreover, QS signaling is interfered with by an organism, which does not produce QS signals, but takes advantage of QQ processes [166]. Several eukaryotes, including plants, animals, and hosts of QS-emitting pathogens, express enzymes that can inactivate QS signals [106]. As mentioned above, several studies evaluated the implication of highly conserved paraoxonases (PON1, PON2, and PON3) [173,215,216], for instance, in defense against the pathogen *P. aeruginosa*. The serum and tracheal epithelial cells of mammals could efficiently inactivate long-chain AHLs of this pathogen [196,197]. Most natural environments harbor diverse microorganisms, and within these communities, bacteria compete with their neighbors for space and resources. Therefore, competitors also evolved several mechanisms to disarm QS systems to avoid bacterial colonization and competence. Inhibitors and antagonists of signal reception [166,217] or enzymatic inactivation were identified among bacteria in natural environments, as already pointed out above [166,218,219,220]. However, the number of models investigated under natural conditions is low, and efforts towards deciphering QQ functions in a rational biological context at the cell-, population-, microbiota- and metaorganism level are underrepresented. 

Investigations on QQ also extended to applied domains for developing antibacterial and antidisease strategies that target pathogens and biofilm-forming bacteria in medicine, agronomy, and industry [221]. The development of treatments based on QS interference is largely driven by alternative or complementary approaches to often ineffective antibiotics [222,223,224]. Conceivable biotechnological applications are manifold, and several examples are published to date, but their application is still only a potential, and studies have to be conducted to direct the potential to real use. In aquaculture, effective alternatives to antibiotics are urgently needed since it is significantly affected by disease outbreaks of often antibiotic-resistant pathogens [225]. Counteracting resistance development, antibiotic administration gets more and more restricted in aquaculture. Known opportunistic pathogens such as *Vibrio* sp., *Aeromonas sp.*, and *Pseudomonas* sp. often regulate pathogenesis through QS; consequently, disruption of QS as a new anti-infective approach has great potential for application in aquaculture [226]. Brominated furanones were found to be effective in neutralizing the growth retarding effect of *V. harveyi* strains. They improved the survival and growth of rotifers [227,228] and further protected brine shrimp *Artemia franciscana* and rainbow trout *Oncorhynchus mykiss* from pathogenic *Vibrio* spp. infections [229,230]. Several bacterial metabolites were also able to block QS-regulated phenotypes in aquaculture pathogens, among those *Shewanella* sp. [231,232], *Halobacillus salinus* [217], and various gut bacteria from marine eukaryotes [233]. By incorporating kojic acid from *Aspergillus* spp. into a nontoxic paint, bacterial and diatoms colonization and growth were successfully controlled by QS interference in artificial marine settings [234]. In industry, QQ is applied in wastewater treatment, where membrane bioreactors used for reclamation and desalination of brackish and seawater are constrained by biofouling of the membrane filters [235]. Fouling biofilms formed by *Aeromonas hydrophila* and *P. putida* [236] are prevented by small-signal interfering compounds added to antifouling coatings or by immobilizing QQ enzymes or marine organisms engineered to secrete QS inhibitors [237,238,239,240]. QQ also finds the way into plant cultivation. Epiphytic bacteria are exploited for controlling diseases by interfering with the QS-regulated virulence of plant pathogens like *P. syringae* [241,242]. Engineering the production of QQ enzymes into plants and plant-associated microbes is expected to help crop protection, as already demonstrated for transgenic tobacco and potato plants, which heterologously expressed Aiia lactonase, resulting in reduced pathogenicity of *Erwinia carotovora* [190,191]. Finally, QQ is also contemplated to be applied in medicine. The use of garlic as a QS inhibitor against *P. aeruginosa*, which is intrinsically resistant to many antibiotics and causes chronic infections, was demonstrated by Rasmussen et al. [243]. This treatment made the biofilm susceptible to antibiotics, such as tobramycin, and phagocytosis by neutrophils [244]. The first clinical trial on the usage of garlic oil macerate as a QS inhibitor for treating human patients suffering from cystic fibrosis was reported in 2010 [245] and resulted in a slight improvement of lung function, weight, and symptoms score of cystic fibrosis patients. Moreover, Hentzer et al. demonstrated that biofilm formation and virulence factor production in *P. aeruginosa* was reduced in the presence of synthetic furanones, which have the potential to be incorporated as QS inhibitors on the surfaces of surgical implants and catheters [240,246]. However, they later showed toxic side effects [228]. 

One of the most important prerequisites to finding effective QQ compounds is their detection with biosensors. AHL and AI-2 reporters were developed throughout by different researchers based on fusing a QS-controlled promoter to a reporter gene [128,144,184,236,247,248,249,250,251,252,253,254]. These biosensors allow for sensitive, quantitative, and real-time detection of QS signals. The reporter strains cannot often produce native QS signals; however, they can respond to exogenous autoinducers, often with a detectable phenotype, such as violacein pigment production in *Chromobacterium violaceum* CV026 [249] and bioluminescence production in *V. harveyi* [255] or *A. tumefaciens* A136 [256]. Most reporters were initially designed to identify new signaling molecules. The QS-promoter is induced by signal molecules possibly present in the environment, leading to the expression of the respective phenotype. By simultaneous addition of defined amounts of promoter-inducing autoinducers in the assay, these biosensors can also be used to identify QQ compounds, interfering with these signal molecules. In brief, they mimic the natural QS system with easily identifiable phenotypes [257]. Remarkably, there are mostly AHL-QS-based reporters published, which allow the identification of AHL-interfering compounds such as AHL-degrading [218] or -modifying [203] compounds as well as AHL agonists [258] and antagonists [259]. *V. harveyi*, a reporter for detecting QQ compounds against Gram-negative and interspecies-specific QS, was developed to identify potential QQ active compounds by the absence of luminescence [260]. Based on this type of system, a screen will indicate a QS-interfering compound by the disappearance of the reporter signal. One crucial problem of this procedure is that factors other than QQ compounds can also cause a reduction in the signal, e.g., by reducing cell growth. Thus, it can be difficult to obtain reliable information regarding the specificity of a QS-interfering compound that shows additional pleiotropic effects because the decrease in reporter signal is not necessarily proportional to the decrease in monitored out read [243]. Rasmussen et al. designed another type of screening system termed Quorum Sensing Inhibitor Selector (QSIS) to circumvent these problems. The QSIS system is based on *E. coli*, which comprises an AHL-inducible lethal gene encoding a toxic protein. When the strain senses AHLs in the surrounding environment, the lethal gene is expressed and consequently growth will be inhibited. In contrast, the presence of a QS-interfering compound rescues the bacteria, since expression of the lethal gene is not induced and the bacteria are able to grow [261]. This method of positive selection for growth has proven powerful for isolation of both, synthetic compounds and extracts of plants and fungi with AHL-quenching activities [98]. As already mentioned, in contrast to AHLs, only a few AI-2 QQ compounds were identified to date, probably due to the lack of appropriate reporter systems [165,262]. A few reporter systems for the detection of AI-2 like compounds can reportedly be used, in principle, for identification of AI-2 quenching activities. One example is the above mentioned *V. harveyi*-based reporter system with a mutated autoinducer synthase (LuxS) that can be used to detect external accumulation of AI-2, leading to bioluminescence [263]. A second reporter system is based on *lacZ* fusion to the *E. coli* AI-2 inducible promoter *lsrA* [264]. Moreover, reporter strain AI2-QQ.1 was established based on the innovative strategy of Rasmussen and collaborators, which now allows identifying for novel, nontoxic biomolecules interfering with AI-2-based QS using positive selection [254]. Nowadays, bacterial isolates, extracts, and metagenomic and synthetic libraries can be rapidly screened for QQ compounds with such biosensors, and the compounds further tested for their application. However, finding new QQ strategies and their effective application in controlling pathogens and bacterial biofilms raised questions about the potential for resistance development against QQ agents. This has become a controversial discussion [265]. The lack of direct effects on the viability of bacteria resulted in the hypothesis that selection for and the appearance of resistant mutants might be less frequent than traditional antibiotic treatment. However, Defoirdt et al. controverted the assumption that QS disruption is not leading to resistance and suggested that the fitness of bacteria can be affected through variability in QS core genes [266]. Additionally, a study has demonstrated that QQ compounds can indeed generate QQ resistance in *P. aeruginosa* [267]. Bacteria could easily escape from QQ approaches by altering the expression of core genes in the targeted QS signaling pathway, such as the genes involved in the signal synthesis, detection, and transduction. It is thus likely that bacteria may develop resistance to QQ; however, it was suggested that the chances of developing resistance are smaller than for conventional antibiotics [225].

In conclusion, QQ strategies have evolved in many pro-and eukaryotes as a mechanism for recycling or clearing their own synthesized QS signals or as a competitive strategy against QS signal-producing organisms. Moreover, QQ might become an effective alternative to combat infections and bacterial biofilms, either as single agents or in combination with antibiotics or other alternative strategies. However, future studies should focus on the underlying QQ mechanisms at a molecular level, their biological role in microbial communities, and their use as antibacterial treatment under realistic conditions to exclude toxic side effects.

### 2.5. Microbe–Host Interactions

Half a century ago, Lynn Margulis first recognized the importance of bacteria in the evolution of higher organisms [268,269,270,271]. In 2007, the term “hologenome” was introduced by Ilana Zilber–Rosenberg and Eugene Rosenberg to describe the sum of the host genome and associated microbial genomes [272,273,274]. This settled the base for a still ongoing paradigm shift in biology. Nowadays, a new conceptual framework—the holobiont/metaorganism concept—is established, considering a holobiont/metaorganism as the sum of a multicellular host and its associated species (as illustrated in Figure 7) [275]. The holobiont theory considers all associated species irrespective of their type of association (transient or permanent) or their function. The metaorganism concept focuses on the function and contribution (beneficial or detrimental) of the host-associated microbiota in a given environment, which depend on the identity, abundance, and activity of the microbes (in the following, the term “metaorganism” is primarily used) [272,274,276]. The growing awareness that multicellular organisms cannot be considered in isolation but only in the interdependence with their associated microbes led to two important insights. Firstly, the health and fitness of a host appear to be fundamentally multiorganismal, where any disturbance within the complex partnership can have drastic consequences for the members’ health. Secondly, host and microbes intensively interact and coevolve [277,278,279] (as illustrated in Figure 7). Several studies already indicate that a specific host-associated microbiota contributes to host metabolism, development, organ morphogenesis, pathogen protection and immunity, behavior, environmental sensing and adaptation, developmental transitions, and reproduction [278,280,281,282,283,284,285,286,287,288,289,290,291,292,293,294] (as illustrated in Figure 7). A specific microbe or consortia function is not static because it depends on the host’s developmental stage, age, reproductive state, or physiological condition [276,278,295]. The vast contribution of the microbiota to the fitness of its host was unraveled mainly in the last decade. Protection against pathogens and provision of essential nutrients were identified as the most general and important contributions of the associated microbiota to the host´s health. In corals, commensal bacteria protect against the bleaching pathogen *V. shiloi* [296] and produce inhibitors to reduce colonization by pathogens on coral mucus [297]. Several experiments further reported that sterile animals are considerably more sensitive to infection and death following administration of a pathogen than conventional animals, e.g., shown for infection of guinea pigs with *Shigella flexneri* [298], mice with *V. cholera* [299], and rabbits with *Bacteroides vulgatus* [300]. Likewise, in humans, the normal microbiota was shown to protect against infection by pathogens in the oral cavity, the intestine, the skin, and the vaginal epithelium [301,302,303,304]. Representatives of the genera *Pseudomonas*, *Flavobacteria*, and *Bacillus* can protect plants from phytopathogens through direct interaction with the pathogen or by inducing systemic resistance in plant hosts [305,306]. In general, microorganisms residing in a plant’s rhizosphere are essential for plant growth promotion, disease suppression, removing toxic compounds, and assimilating nutrients to plants [307]. Moreover, utilizing such beneficial microbes for crop productivity presents an efficient way to modulate the crop yield and productivity by maintaining the health status and quality of the plants [308,309,310]. The utilization of beneficial microbe-plant interactions is now turning into the standard against the chemical-based and synthetic pesticides and fertilizers in the agriculture industry [307,311,312]. Moreover, the contribution of the microbiota to the nutrition of the hosts is known for many years. For example, chemoautotrophic bacterial symbionts, like sulfur-oxidizing bacteria, synthesize organic matter from CO_2_ and are the primary source of nutrition for their animal host [313]. In turn, the host, like clams and mussels, provides its symbionts a habitat, in which they have access to the substrates of chemoautotrophy (O_2_, CO_2_, and reduced inorganic compounds such as H_2_S) [314]. In humans, the gut microbiota is a complex ecosystem that plays an essential role in the catabolism of dietary fiber, production of vitamins and amino acids, and detoxification of harmful chemicals [315,316]. In recent years, several studies further demonstrated that the microbiome also contributes to the development of a variety of tissues, functions, and organs [273,317]. An illustration of bacteria-dependent development can be seen for many green algae, which develop abnormally in the absence of bacteria [318]. For example, the marine green alga *Ulva lactuca* loses its typical leafy morphology in axenic culture and develops into pincushion-like colonies. However, these abnormal algal colonies can be restored to their typical morphology by recolonization with appropriate marine bacteria [319]. Also, gut bacteria were shown to shape the tissues, cells, and molecular profile of the mammalian gastrointestinal immune system during development [320]. Microbes interact with the host cells through adhesive molecules on their surface, thus promoting interaction with host cell receptors and triggering host responses among those immune responses and metabolic and behavioral reactions [321]. For instance, first experiments with mice demonstrated that gut microbiota affects the brain and, consequently, behavior [322]. It is most likely that during evolution, gut microbiota colonization became integrated into the programming of brain development affecting the central nervous system and behavior [323]. Bacteria communicate with the brain via changing levels of dietary metabolites and hormones [324]. The gut microbiota can be a key regulator of mood, cognition, pain, and obesity [325]. Understanding the microbiota–gut–brain axis can give new insights into individual variations in cognition, personality, mood, sleep, and eating behavior and how microbes contribute to a range of neuropsychiatric diseases ranging from affective disorders to autism and schizophrenia. Understanding such complex host-microbe interactions can thus be a key for developing therapeutics against diseases, but also finding pre- and probiotics to prevent infections and disorders in the future [326]. Besides cooperation between microbiota and host, there are numerous examples of interactions among different species of microorganisms within a metaorganism, like cross-feeding between microbes in the mucus of coral holobionts [327] or sharing of “public goods” in microbial biofilms on the outer surface of animals or plants as well as the digestive tract of animals [328], ultimately resulting in a fitness benefit for the metaorganism as a whole. Some cooperative microbial interactions are so called byproduct mutualistic interactions. For instance, the host benefits from byproducts of a bacterial metabolism as known for short-chain fatty acids produced by bacteria in the large intestine of mammals during anaerobic metabolism [329,330]. The aphid-*Buchnera* symbiosis exemplifies that benefits for the host in the end also possess an advantage for the symbiont, and thus for the whole metaorganism, since energy-costly overproduction of amino acids by the endosymbiont, which are essential to the insect host, finally configure the environment for the endosymbiont [331,332]. The manifold examples of cooperation do not contradict the competition existing within the metaorganism among different microbial community members, as discussed above (see Section 2.4).

The metaorganism concept leads to the assumption that the evolution of multicellular host organisms primarily occurred by cooperation with their associated microbiota. Cooperation among microbes and between microbes and their host coevolved over millions of years, exemplified by arbuscular mycorrhizal fungi and roots of vascular plants (ca. 400 million years ago) [333]. Here, fungi of the phylum Glomeromycota penetrate the cortical cells of the roots of a vascular plant forming arbuscules. The fungus enables its plant host to capture nutrients such as phosphorus, sulfur, nitrogen, and micronutrients from the soil. The arbuscular mycorrhizal symbiosis most likely played a crucial role in the evolution of vascular plants and their initial colonization on land. Arbuscular mycorrhizal symbiosis is presumed to be the most prevalent known plant symbiosis and is found in 80% of vascular plant families [334]. Evolved mutualistic relationships are further known for endosymbiotic algae and corals (ca. 240 million years ago) [335], ruminants and their microbiota (ca. 60 million years ago) [336], and the great apes and gut microbiota (15–20 million years ago) [337,338]. The gut microbiome also demonstrates evolution by acquiring a microbiota in ruminants and termites, where the microbes degrade cellulose of plant material, enabling the host to metabolize those nutrients [339,340]. One of the major advantages of microbiota acquisition is that it allows for rapid adaptation to new environments. Such an example is shown for the bacterium *Burkholderia,* which is resistant to an insecticide, and further led to the resistance of the host insect to the insecticide [341]. The evolution of metaorganisms can also be acquired by the horizontal transfer of microbial genetic information to the host chromosome [342]. There are numerous known examples of horizontal gene transfer (HGT) between symbionts and their hosts, e.g., transfer of biosynthetic genes for carotenoid production from a fungus to aphids [343], gene transfer from the endosymbiont *Wolbachia* to the arthropod host [344], and transfer of the long interspersed nuclear element (LINE-1) from human to the pathogen *Neisseria gonorrhoeae* [345].

In summary, all animals and plants are inhabited by microbial organisms, which influence the health and fitness of their hosts, ultimately forming a metaorganism harboring complex interactions among microbial community members and between the microbes and their host. Research on host–microbe interactions became an emerging cross-disciplinary field. A diverse and complex microbiome confers immunological, metabolic, and behavioral benefits; its disturbance can contribute to disease development. However, the molecular and cellular mechanisms controlling interactions within the metaorganisms are poorly understood, and many key interactions between the associated organisms remain unknown. Future studies should focus particularly on the functional consequences of the interactions and the impact of the microbiota on the host’s life history and evolutionary fitness.

## 3. Methods for Studying Microbial Interactions

Microorganisms play a vital role in various ecosystems, and characterizing interactions between them is an essential step towards understanding the organization and function of microbial communities [346]. A key for understanding microbial interactions is the continual development of cutting-edge methods and controlled experimental platforms. The fact that the vast majority (95–99%) of microbes were not cultivated points to the urgent need for cultivation-independent approaches and methods, which are summarized as “metagenomics” [347]. The term metagenomics was first used by Jo Handelsman in 1998 [348]. Metagenomics is defined as the genomic analysis of an assemblage of organisms in a given habitat by direct extraction and cloning of the whole DNA [349]. Consequently, metagenomics can unlock the massive uncultured microbial diversity present in an environment. Metagenomics studies identified many novel microbial genes coding for metabolic pathways, such as energy acquisition, carbon and nitrogen metabolism, and novel genes, particles, and compounds applicable to biotechnology [350,351,352,353,354]. Metagenomics allows the investigation of microbes in their natural environments and complex communities associated with abiotic and biotic surfaces as in metaorganisms [355]. There are two basic types of metagenomics studies: (i) sequence-based metagenomics that involves sequencing and analysis of DNA from environmental samples, and (ii) function-based metagenomics that comprise screening for a particular function or activity [356]. Sequence-based metagenomics can be used to assemble genomes, identify genes, find complete metabolic pathways, and analyze microbial diversity and abundance, providing information about the ecology of the microbes in the community and within a metaorganism [357,358,359,360,361,362]. Functional metagenomics is a powerful experimental approach for studying gene function, starting from the extracted DNA of mixed microbial populations or a whole metaorganism. A functional approach relies on the construction and screening of metagenomic libraries-physical libraries that contain DNA cloned from environmental metagenomes into vector backbones. Cosmid- or fosmid-based libraries are often used due to their large and consistent insert size and high cloning efficiency [350,351,352,353,356,363]. Moreover, metagenomics led to the discovery and characterization of a wide range of biocatalysts and novel compounds for clinical, industrial, and biotechnological applications. For instance, the investigation of symbiotic bacteria by metagenomics led to the identification of a rare type I polyketide synthase, allowing the generation of novel antitumor compounds [364]. Analysis of metagenomic libraries revealed a high frequency of novel antibiotics present in soil [365]. A novel antibiotic, palmitoylputrescine, was found in the tropical plant (bromeliads) tank water, and a novel bacterial mechanism for inactivation of tetracycline was found in the oral metagenome [366]. Metagenomics further identified novel xenobiotic degradation pathways used by prokaryotes in the environment [367], which can be used in biotechnology. One of the main areas of metagenomic research from the beginning was discovering novel biocatalysts, including esterases, nitrile hydratases, alcohol reductases, amidases, cellulases, amylases, glycogen-branching enzymes, and pectate lyases [368,369,370,371,372,373,374]. 

Besides metagenomics, advances in sequencing and “omic” technologies improved effectively tracking microbial community composition and the metabolic activity of microbes, further allowing for correlations on microbial functions [375]. Sequencing RNA from both prokaryotic and eukaryotic cells simultaneously can reveal how the host and its associated microbiota interact at the gene expression level [376]. Additional in situ methods can also help to capture microbial interactions [377]. Techniques for quantitative imaging of labeled bacteria and their surroundings, including fluorescence in situ hybridization labeling of bacteria and noninvasive imaging of extracellular milieu components, add a critical spatial dimension to microbial studies [377,378]. Metabolic labeling enables even tracking microbial activity [379]. Nowadays, synthetic approaches also help to explore the complexity of microbial interactions [380,381]. On the one hand, synthetic microbial communities offer reduced complexity and can more easily be mathematically modeled [382]. On the other hand, advances in synthetic biology enabled the engineering of microbes with genetically defined properties [383]. These engineered bacteria were combined with artificial environments to study microbial interactions in response to environmental cues. Experimentally, microchamber-based methods were developed to mimic the natural environment, exemplified in a microfluidic assay to assess dynamic root–microbe interactions [384] and in an in situ chemotaxis assay (ISCA) to study marine microbial behaviors at spatially relevant scales [385]. However, it should not be neglected that understanding microbial community interactions, particularly all the complex interactions that take place in metaorganisms, also rely on cultivation-dependent approaches [295]. The cultivation of bacteria is highly biased toward a few phylogenetic groups. New cultivation concepts were and will still be developed based on an improved understanding of the ecology of previously uncultivable bacteria. Here, media that mimic the natural types and concentrations of substrates and nutrients, high-throughput cultivation techniques, and approaches that exploit biofilm formation and bacterial interactions crucially improved cultivation techniques [386]. Metagenomics and single-cell genomics can further reveal unknown metabolic features, the information needed for improved cultivation or even cocultivation of microbes [387]. Notably, studies on host–microbe interactions rely on the combination of largely (gnotobiotic) or completely (axenic) germ-free hosts with cultured microbial isolates to gain detailed insights into the contributions of microbes to metaorganism function [284,388]. Such recolonization experiments enable assigning a function to specific microbes [276,388,389] as well as to determine the colonization dynamics of microbes [388,390] and to elucidate bacteria–bacteria interactions [284,391]. All the presented technologies can provide insights into the complexities of microbial communities, the function of microbes, and the interaction among these microbes with their hosts. Despite the remarkable advances in the study of microbial communities, we are still far from understanding the complete picture. Moving beyond purely observational approaches, experimental and computational methods that facilitate the interpretation of microbe interactions with each other and their communities must be enhanced. Computational modeling and analysis tools need to be developed and implemented to investigate environmental correlations and to understand microbial dependencies and their coevolution [392].

Besides understanding microbial interactions, research also focused on investigations to use microbial consortia in biotechnology processes, including fermentation, waste treatment, and agriculture, for millennia [393]. However, only an improved understanding of natural microbial ecosystems and the development of new tools to construct synthetic and engineered consortia vastly expanded the possibilities of using microbial consortia for diverse applications, including bioproduction of medicines, biofuels, and biomaterials [394]. Microbial communities comprising several partners can often jointly perform complex processes more efficiently, yielding the desired product at an increased rate than that of a single species [395]. Interactions between the microbial partners in these mixed communities are expected to significantly impact the microorganisms’ combined performance and the bioprocess as a whole. Commensal or mutualistic interactions among microbial members of a consortium can significantly enhance the product outcome of the bioprocess, ensuring their industrial application and long-term stability [395]. Beyond being simply positive or negative, beneficial or inhibitory, microbial interactions can involve a diverse set of mechanisms, dependencies, and dynamical properties. This complexity of interactions must be studied by further developing cutting-edge techniques that enable the elucidation of multidimensional and dynamic relationships among microbes and between microbes and the host.

## 4. Conclusions

Microorganisms live in close contact with each other and to multicellular hosts, usually including many species. Additionally, microbes are exposed to variations in the environment, which in turn affect the interactions. Microbial interactions are thus highly complex, and many mechanisms and molecules are involved [5]. Studies on microbial interactions led to significant findings in microbiology, botany, zoology, and ecology. Research on microbial interactions also enabled discoveries for clinical, industrial, and biotechnological applications, e.g., antimicrobial drug development based on natural products like QS interfering compounds. Further, the realization that a germ-free multicellular organism does not exist in nature led to the holobiont/metaorganism concept and a paradigm shift in life science. It is now believed that it would be impossible to fully understand a multicellular organism without considering its associated microbes, but it is possible to study microbes without knowledge of animals and plants. There is still a lot to understand about the molecular mechanisms and languages used by microorganisms and the molecules and signals involved in microbial interactions, in particular with the host. The development and refinement of tools and methods, including in vitro and in vivo models, are urgently needed to understand better and characterize microbial interactions with more molecular details. Elucidating complex microbial interactions in an ever-changing environment in nature is probably the most challenging endeavor.

## Figures and Tables

**Figure 1 biology-10-00496-f001:**
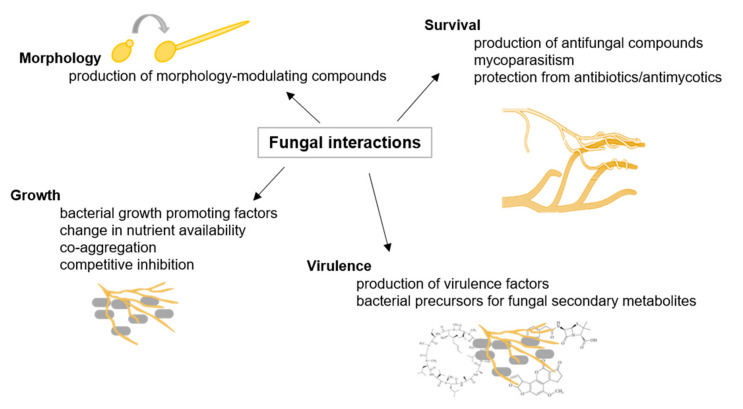
Fungal interactions. Examples of fungal interactions are illustrated with their respective consequences, adapted from [36].

**Figure 2 biology-10-00496-f002:**
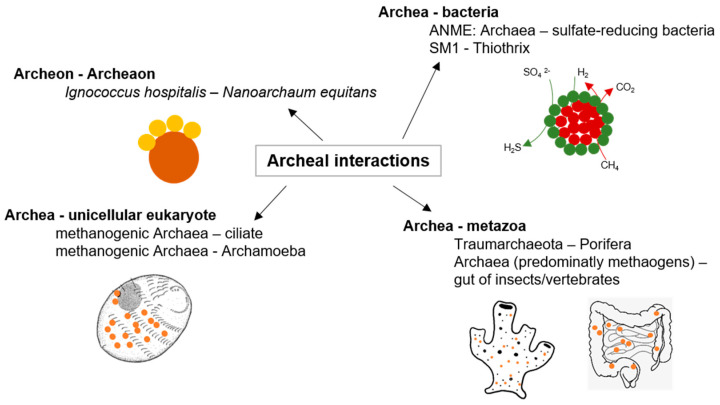
Archaeal interactions; examples of archaeal interactions adapted from [38].

**Figure 3 biology-10-00496-f003:**
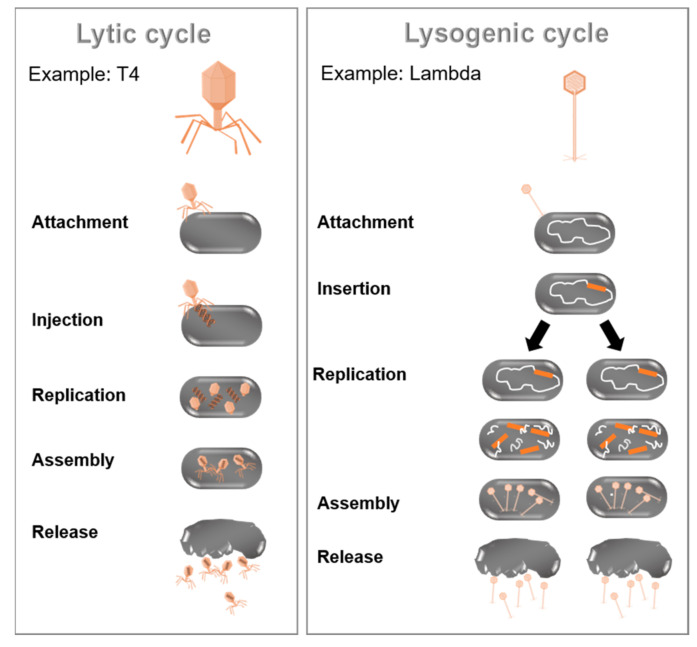
Phage proliferation. (**Left**) Infection by lytic phages like *E. coli* phage T4 leads to the multiplication of virus and subsequent lysis of bacterial host cell. (**Right**) In lysogenic cycle, phage genome (e.g., of *E. coli* phage Lambda) is integrated into bacterial chromosome, and an inactive prophage is replicated as part of host chromosome. Environmental triggers cause excision of phage genome and entry into lytic cycle.

**Figure 4 biology-10-00496-f004:**
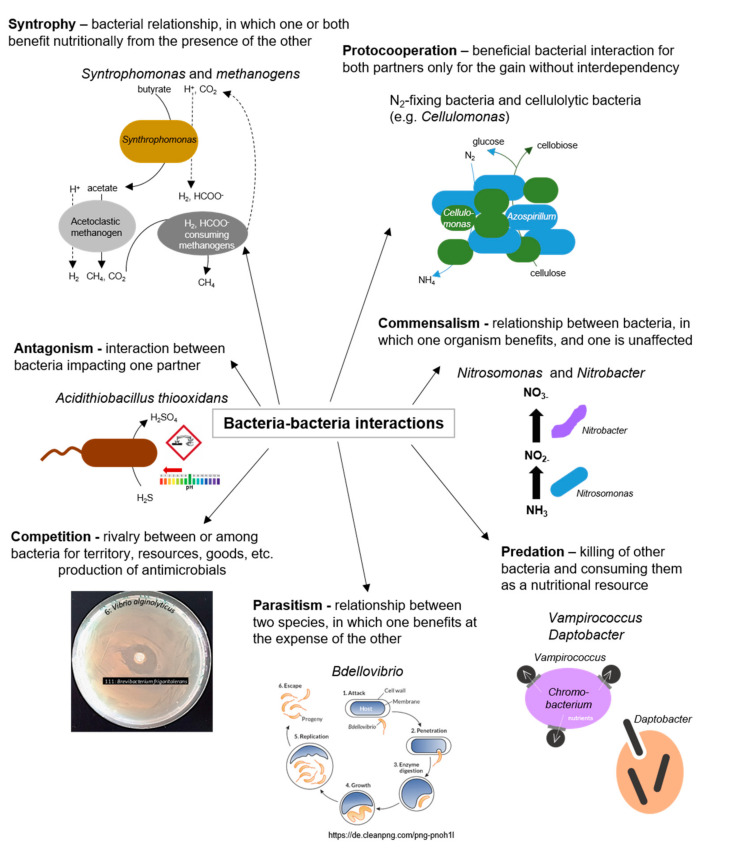
Bacteria–bacteria interactions; examples of positive and negative interactions among bacterial species.

**Figure 5 biology-10-00496-f005:**
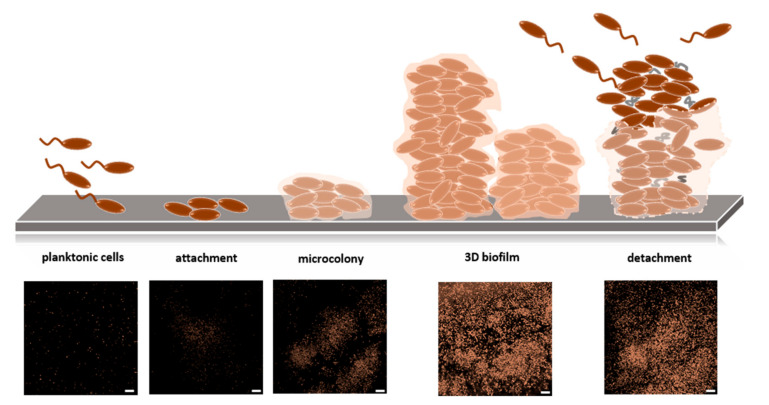
Biofilm development. (Upper panel) Free-swimming bacteria initially attach to a solid surface, and colonizing bacteria further form structured aggregates called microcolonies. Biofilms are composed of numerous microcolonies, which are encased in an extracellular polymeric matrix. Biofilms permanently undergo composition/decomposition. (Lower panel) Confocal Laser Scanning Micrographs of *Klebsiella oxytoca* M5aI biofilm formation.

**Figure 6 biology-10-00496-f006:**
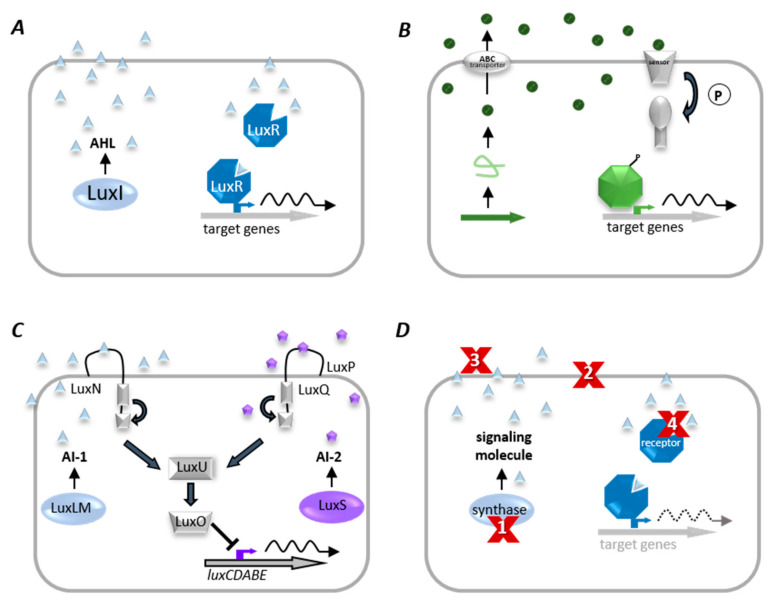
Quorum sensing (QS) systems and Quorum quenching (QQ) strategies. (**A**) Gram-negative bacteria produce diffusible autoinducers (AI, triangles) by a LuxI homologous synthase. AIs diffuse into the cell and bind to cognate receptor (LuxR homolog). This complex binds at target gene promoters and activates their transcription. (**B**) A precursor peptide (loop) is produced and modified (circles) by Gram-positive bacteria and then secreted via an ATP-binding cassette (ABC). A two-component system detects signaling molecules, and phosphorylated response protein binds to specific promoter genes to modulate their expression. (**C**) The QS system of *V. harveyi* combines Gram-negative and Gram-positive QS elements, in which acyl-homoserine lactones (AI-1, triangles) are synthesized by LuxLM, and a second universal AI (AI-2, pentagons) is synthesized by the enzyme LuxS. AIs are detected by two-component systems whose signals are transduced by phosphorelay and end in the expression of the luciferase structural operon (*luxCDABE*). (**D**) Examples of QQ strategies. (1) inhibition of AI biosynthesis; (2) inhibition of signal transport; (3) degradation, modification, or antagonism of AIs, and (4) inhibition of signal recognition.

**Figure 7 biology-10-00496-f007:**
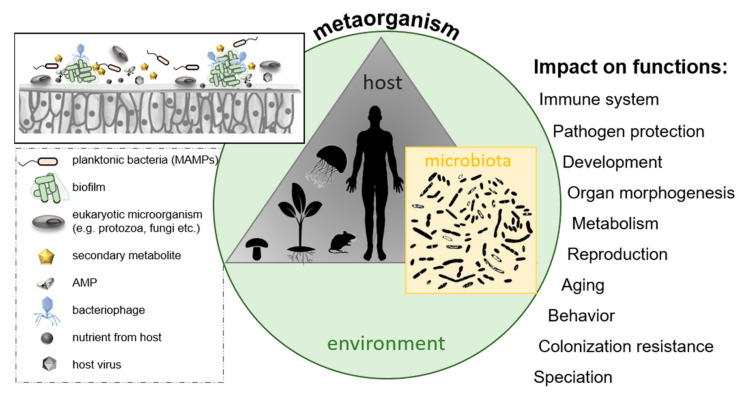
The metaorganism concept; a metaorganism consists of a multicellular host and its associated microbiota located in a specific environment. Selected factors are presented that might influence bacterial colonization of host surfaces, ultimately affecting various host functions.

**Table 1 biology-10-00496-t001:** Types of microbial interactions adapted from [16].

Interaction	Characteristic	Species A	Species B	Example
Mutualism	Symbiosis needed for survival in a specific habitat	Benefits	Benefits	Root nodules [6]
Synergism	Another improves the growth of one partner	Benefits	Benefits	Crossfeeding of acetate between bacteria [17]
Commensalism	One partner benefits and the other is not harmed nor improved	Benefits	Not affected	Nitrification with *Nitrosomonas* and *Nitrobacter* [18]
Parasitism	Host is compromised	Benefits	Harmed	*Bdellovibrio sp.* and BALO require Gram-negative bacterium for growth [19]
Competition	Rivalry for space and nutrients	Harmed	Harmed	Soil bacteria compete with fungi for nutrients [20]
Antagonism	Product(s) of one partner impact another	Not affected or benefits	Harmed	Production of antibiotics [21]

## Data Availability

Not applicable.

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
