# Peer review of "Friends or Foes—Microbial Interactions in Nature"

_biology, 2021, doi:10.3390/biology10060496_

Round 1
Reviewer 1 Report
In this paper, the author presented many interesting aspects of the dual roles of microbial interactions.
This manuscript contains much information but some of them are too descriptive without specific contribution or specific effects. The role of microbial interaction in agriculture is not discussed well. This is an important area. I suggest adding more literature.
I think only an abstract is fine. No need to present a Simple Summary and abstract. Please use only one.
Please make a separate 'Introduction' section.
In the Table, each sentence should be capitalized properly. The first letter of the first word should be capitalized.
References are not formatted as per MDPI guidelines.
Author Response
Point-to-point response
Reviewer 1
In this paper, the author presented many interesting aspects of the dual roles of microbial interactions.
This manuscript contains much information but some of them are too descriptive without specific contribution or specific effects. The role of microbial interaction in agriculture is not discussed well. This is an important area. I suggest adding more literature.
Response:
Dear Reviewer,
Thank you for the time and willingness to review the manuscript to conclude that “many interesting aspects of the dual roles of microbial interactions” are presented in the Review paper. Please find below the point-to-point response written in regular font for your thoughtful recommendations presented in italic font.
This manuscript contains much information but some of them are too descriptive without specific contribution or specific effects. The role of microbial interaction in agriculture is not discussed well. This is an important area. I suggest adding more literature.
I thank the reviewer for the meaningful comment. The review does not intend to give a complete picture about microbial interactions and should more serve as a general overview of microbial interactions in nature. Cooperative and competitive interactions within different microbial phyla are in the focus of this review, emphasizing microbial cell-cell communication (quorum sensing) and metagenomics. I completely agree with the reviewer that the role of microbial interactions is crucial in agriculture. According to the reviewer's comment, I added information on microbial interactions in agriculture in the lines 134 – 141 and 842-849 (revised manuscript version without track changes), respectively with respective references.
I think only an abstract is fine. No need to present a Simple Summary and abstract. Please use only one.
Thank you for the comment on only presenting an abstract. However, the guidelines of MDPI journal Biology request both an abstract and a simple summary. Therefore, I decided to stick to the journal requirements to present an abstract and a simple summary as requested in the word template for submission.
Please make a separate 'Introduction' section.
I thank the reviewer for the well-taken comment. The first paragraph now sounds “Introduction – microbial interactions at a glance” in the edited manuscript.
In the Table, each sentence should be capitalized properly. The first letter of the first word should be capitalized.
I apologize for this formatting mistake. I capitalized the first letter of each first word as recommended by the reviewer.
References are not formatted as per MDPI guidelines.
I apologize for any mistakes in the reference list by using the wrong formatting guidelines. References were checked and revised using the newest version of MDPI - Chicago Endnote Output Style.
Reviewer 2 Report
This is a well written review article that covers complex interactions between a variety of microorganisms including bacteria, archaea, fungi, microalgae and viruses. The review is thorough and well referenced. The figures enhance the review and help in overall understanding of the content. Only minor revisions are requested by this reviewer.
General comment: some inconsistency with either capitalization or not of "archaea"
"Quorum Sensing (QS)" is defined lines 279, 292, 401. This only needs to be done once.
After defining QS, use QS instead of spelling it out such as lines 298, 381, 768 and 1048. Please make consistent
“Quorum Quenching” (QQ) is also defined twice: lines 301 and 567
Specific comments:
Line 125: Expand the genus name in S. coelicolor
Line 167: Should converting be "convert"?
Line 563: S. aureus should be in italics
Line 643: P. aeruginosa should be in italics
Line 922: define LINE-1 prior to using acronym
Author Response
Reviewer 2
This is a well written review article that covers complex interactions between a variety of microorganisms including bacteria, archaea, fungi, microalgae and viruses. The review is thorough and well referenced. The figures enhance the review and help in overall understanding of the content. Only minor revisions are requested by this reviewer.
Response:
Dear Reviewer,
Thank you for your time and willingness to review the manuscript and the entirely positive reaction to the Review paper. I am happy to respond to the minor comments below.
General comment: some inconsistency with either capitalization or not of "archaea"
I thank the reviewer for pointing to this formatting mistake. The word “archaea” is not anymore capitalized throughout the manuscript, according to the comment.
"Quorum Sensing (QS)" is defined lines 279, 292, 401. This only needs to be done once. After defining QS, use QS instead of spelling it out such as lines 298, 381, 768 and 1048. Please make consistent
I apologize for the many introductions of the abbreviation QS for Quorum sensing. The edited manuscript is now consistent in terms of the abbreviation usage as suggested by the reviewer.
“Quorum Quenching” (QQ) is also defined twice: lines 301 and 567
I apologize for the mistake and revised it as suggested by the reviewer.
Specific comments:
Line 125: Expand the genus name in S. coelicolor
Line 167: Should converting be "convert"?
Line 563: S. aureus should be in italics
Line 643: P. aeruginosa should be in italics
Line 922: define LINE-1 prior to using acronym
All meaningful specific comments of the reviewer were taken into account and revised accordingly.
Reviewer 3 Report
Nice review but would benefit from an overhaul of grammar and English presentation.
Author Response
Reviewer 3
Nice review but would benefit from an overhaul of grammar and English presentation.
Response:
Dear Reviewer,
Thank you for the time and willingness to review the manuscript and rating the submitted Review paper as "nice". I also thank you for the recommendation to optimize the grammar and English presentation of the manuscript since the other reviewers just announced minor spell requirements. However, the complete manuscript was extensively checked and edited for English language and style using the software "Grammarly Premium". A native-speaking colleague additionally edited the manuscript. Please have a look at the revised manuscript, which hopefully now reaches the requirements.
Round 2
Reviewer 3 Report
Draft is improved!